## Perspective

 

**Subject Area:**
developmental biology/genetics/cellular biology

Jun N-terminal kinase, apoptosis, cell competition, tumorigenesis, regeneration

**Author for correspondence:**
Ginés Morata
e-mail: gmorata@cbm.csic.es

# Pro-apoptotic and pro-proliferation functions of the JNK pathway of *Drosophila*: roles in cell competition, tumorigenesis and regeneration

Noelia Pinal, Manuel Calleja and Ginés Morata

Centro de Biología Molecular CSIC-UAM, Madrid, Spain

GM, 0000-0003-3274-5173

The Jun N-terminal kinase (JNK) is a member of the mitogen-activated protein kinase family. It appears to be conserved in all animal species where it regulates important physiological functions involved in apoptosis, cell migration, cell proliferation and regeneration. In this review, we focus on the functions of JNK in *Drosophila* imaginal discs, where it has been reported that it can induce both cell death and cell proliferation. We discuss this apparent paradox in the light of recent findings and propose that the pro-apoptotic and the pro-proliferative functions are intrinsic properties of JNK activity. Whether one function or another is predominant depends on the cellular context.

## 1. Introduction

The Jun N-terminal kinase (JNK) belongs to the mitogen-activated protein kinase (MAPK) family. It performs several functions related to stress-induced apoptosis, cell migration, cell proliferation and regeneration [1–7]. Much effort has been devoted to analyse and discriminate the molecular basis of the different functions of JNK [8–10], especially in vertebrates, where there are three genes (jnk1, jnk2 and jnk3) encoding related JNK proteins [11–13]. A principal feature of the JNK pathway is that it is activated in response to stress or pro-inflammatory cytokines [14–16]. It is of interest that studies on mouse models of JNK function have reported conflicting evidence indicating that JNK may either behave as a pro-tumorigenic or as a tumour-suppressing factor [17,18]. The reasons behind this paradoxical behaviour are not yet fully understood.

By contrast, in *Drosophila*, there is a single JNK protein, encoded by the gene *basket* (*bsk*); the Bsk protein is a substrate for the Hemipterous kinase, which in turn is phosphorylated by other upstream kinases. The latter can be activated by a number of intracellular proteins (see review in [19]). The activation of Bsk leads to the phosphorylation of the transcription factors Jun and Fos that regulate the activity of genes responsible for the various cellular functions associated with JNK activity. It is required for the dorsal closure during late embryogenesis [20,21] and also for the proper fusion of the left and right sides on the midline of the adult cuticle [22]. In addition, a major function of JNK is its involvement in the mechanism of stress response to ionizing irradiation, heat shock, tissue damage, etc. [1,7]. In the case of the imaginal discs (the precursors of the adult cuticular structures), JNK is not expressed during normal development (except in a small zone of the proximal region of thoracic discs), but it is activated to high levels after irradiation or tissue damage [1,7]. The activity of JNK under those conditions triggers the apoptosis programme and the subsequent elimination of the cells expressing the pathway. However, it has been shown that JNK may also exert a pro-proliferation activity: sustained JNK expression is associated with the formation of large tumorous overgrowths in the imaginal discs [7,23–25] and also

with the additional cell proliferation necessary during regeneration processes [1,26,27].

In this report, we focus on events associated with the induction by JNK of the apoptosis programme of *Drosophila* and its implications in cell competition, tumorigenesis and regeneration. First, we argue that JNK has an autocrine function that normally causes cell death. This pro-apoptotic activity is responsible for the killing of cells damaged by irradiation or injury and also of the elimination of viable but out-competed cells during the cell competition phenomenon. Second, we argue that JNK has a paracrine function that induces proliferation of neighbour cells and is responsible for the development of tumours and the regeneration of damaged tissues

## 1.1. Apoptosis in *Drosophila*: pro-apoptotic and pro-proliferative roles of JNK

Programmed cell death (apoptosis) is a process by which cells trigger their own destruction. Ultimately, it involves the activation of several cysteine-proteases, referred to as caspases, which kill the cells by dismantling protein substrates. The execution process by caspases is conserved in all animals, although the initiation events vary among species [19,28–31]. Apoptosis may be developmentally programmed, as in the cases of the inter-digital tissues in vertebrates, the tail structures in amphibians or the joints of *Drosophila* legs [32–34]. There is also non-programmed apoptosis that acts as a response mechanism to stress or other events that may generate damaged or aberrant cells that need to be eliminated [35].

In *Drosophila*, the genetic factors involved in apoptosis are well known (figure 1); it is initiated by the activation of one or more pro-apoptotic genes (*hid, reaper, grim*) that block the activity of the *Drosophila* inhibitor of apoptosis1 protein (encoded by the *diap1* gene). The loss of *diap1* function allows the activation of the caspases and subsequent cell death (see [31] for a detailed review).

We focus here on the wing imaginal disc, in which there is no developmentally regulated apoptosis. However, it exhibits a strong apoptotic response to stress events like ionizing irradiation or tissue damage [1,36]. Wing imaginal cells can also initiate apoptosis as induced by cell competition (see below)

A scheme of stress-induced apoptosis is shown in figure 1. One significant feature of the apoptosis programme of *Drosophila* is that it functions as an amplification loop in which the JNK pathway plays a relevant role. JNK is primarily activated by stress factors, but secondarily also by the apical caspase Dronc ([37], figure 1). This causes a stimulation of the pro-apoptotic role of JNK. This reinforcement of JNK activity is critical for the apoptotic response, because in its absence, the overall levels of the effector caspase activity after stress are much lower [38]. The mechanism by which Dronc activates or stimulates JNK [37,38] is not known.

A principal factor associated with the initial activation of JNK after stress in planarians and vertebrates [39–41] is the appearance of high levels of reactive oxygen species (ROS). Also in *Drosophila*, damage to imaginal tissues causes a burst of ROS production that acts as a trigger of JNK activity [42,43].

In addition to its pro-apoptotic function, the JNK pathway also possesses a seemingly contradictory property; that is the ability to release proliferative signals that can stimulate the growth of the tissue nearby. This paracrine function was

royalsocietypublishing.org/journal/rsob  Open Biol. **9**: 180256

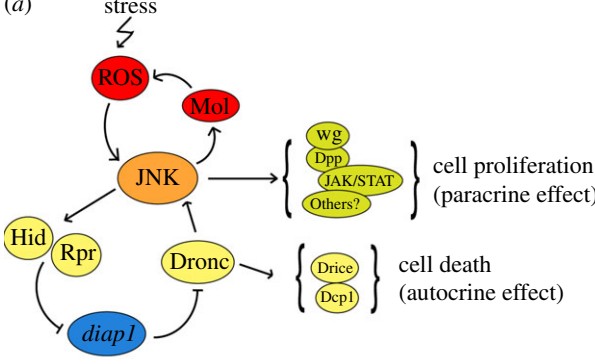

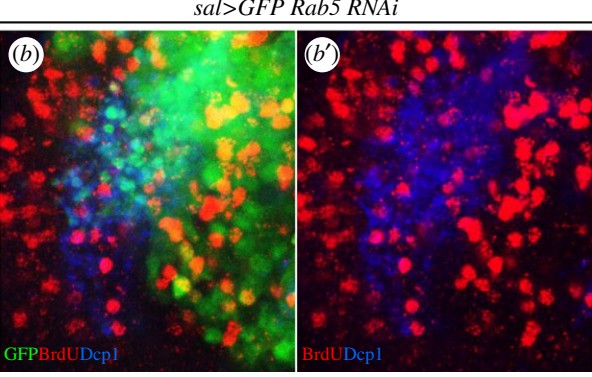

**Figure 1.** Autocrine and paracrine functions of JNK. (*a*) Activation and maintenance of the function of the JNK pathway. After an initiation event (irradiation, heat shock), the high ROS levels produced activate JNK. In turn, JNK activates the pro-apoptotic genes *rpr* and *hid* that suppress the activity of the apoptosis inhibitor *diap1*. The loss of *diap1* function permits the activation of the apical caspase Dronc and subsequently of the effector caspases Drice and Dcp1, which causes the death of JNK-expressing cells; an autocrine effect. The fact that Dronc further stimulates JNK activity results in an amplification loop, necessary for complete apoptotic response to stress. Besides, JNK-expressing cells have the capacity of sending proliferative signals to neighbour cells, a paracrine effect likely achieved by upregulation of other signalling pathways like JAK/STAT, Wg and Dpp. In normal circumstances, the prompt death of JNK-expressing cells makes the proliferative signalling inconsequential, but it may become prominent if the apoptosis machinery is compromised. Besides the stimulation by Dronc, JNK also has the property of self-maintenance, due to a loop generated by the transcriptional activation of *mol*, a DUOX factor that increases the levels of ROS and thus sustains JNK activity. (*b,b′*) Fragment of a sal＞Rab5i GFP wing disc triply labelled with GFP, Dcp1 (blue) and BrdU (red). The GFP cells are defective in Rab5 function and the majority are in apoptosis as indicated by the blue (Dcp1) staining (*b′*). Close to the dying cells, there is an accumulation of BrdU labelled cells, indicating they are actively dividing. The image illustrates both the autocrine (promoting cell death) and paracrine (promoting cell proliferation) functions of JNK.

originally discovered in 'undead' cells: cells in apoptosis in which the presence of the baculovirus protein P35 prevents the activity of the effector caspases and the destruction of the cells, even though they continuously express the apoptosis programme. Undead cells remain alive for the rest of the development and keep secreting mitogenic factors, thus generating hyperplastic overgrowths [7,36,44] (reviewed in [31,45]). This proliferative signalling depends on JNK activity [44] and appears to be mediated by the function of downstream pathways like JAK/STAT, Wg and Dpp [7,44,46,47], although the interactions between these pathways are still unclear. It is also mediated by the downregulation of the Hippo pathway, as indicated by the gain of function of targets of the Yki transcriptional activator [48,49].

The paracrine function of JNK was originally interpreted as a mechanism of compensatory proliferation (i.e. dying cells expressing JNK induce additional proliferation of their neighbours to compensate for their demise). This hypothesis has been questioned by experiments showing that X-ray-damaged tissue can compensate in the absence of the mitogenic signals Wg and Dpp from apoptotic cells [7].

Although it was discovered in cells in which the apoptosis programme was active, the paracrine capacity to stimulate cell proliferation is an intrinsic property of JNK that does not depend on apoptosis. Sustained JNK activity in apoptosis-deficient cells also causes excessive proliferation [7,25].

These two JNK functions, the autocrine induction of apoptosis and the paracrine stimulation of proliferation, occur concomitantly upon JNK activation (figure 1b) [7,25]. The paracrine function is normally inconsequential because the cells in apoptosis emitting mitogenic signals are short-lived. It is only when the apoptotic cells remain alive, like the undead cells, or when cells lack the apoptosis machinery [7,25], that the consequences of the proliferative signalling become apparent.

## 1.2. The autocrine function of JNK is responsible for the elimination of loser cells during cell competition

Cell competition was originally reported [50] in experiments of cell lineage designed to alter the growth rate of marked clones in the imaginal discs of *Drosophila*. The experiments were based on the use of mutations, collectively termed *Minute*, known to cause a developmental delay when in heterozygous condition. The *Minute* genes encode ribosomal proteins [51] and the delay is caused by a slow proliferation rate of heterozygous ($M/+$) cells, presumably due to the limiting amounts of ribosomal proteins in flies that only contain one dose of the gene. The key observation [50] was that although $M/+$ flies are viable, $M/+$ cells are eliminated when in the same population with more rapidly proliferating cells. Subsequent work [52,53] confirmed the observation in different developmental contexts. Later reports [54–56] showed that cell competition also functions to remove cells that are less metabolically active than their neighbours or have different identity.

Cell competition is a context-dependent phenomenon: outcompeted cells (referred to as 'losers') are viable; they are eliminated only when in the same population with cells (referred to as 'winners') that induce their elimination, thus the process relies upon cell interactions. A significant feature is that cell competition appears to function at a very short range [53]; in all the well-characterized cases, the interacting winner and loser cells are very close, and may be in physical contact.

The role of cell competition is not limited to the elimination of cells that are less fit or have inappropriate identity. Importantly, it also functions to eliminate malignant/oncogenic cells that appear in development, thus indicating a tumour-suppressor role [24,57,58].

In broad terms, cell competition behaves as a cell quality control mechanism responsible for the elimination of unwanted cells that are weak, abnormal or malignant. Considering the large number of cells of multicellular animals and the average values of somatic mutation rates, it is clear that the body of animals contain in any moment of their lives a large number of abnormal cells that may compromise the fitness or the survival of the organism. This calls for the existence of a mechanism to remove such unwanted cells (see [59] for a general discussion about the physiological role of cell competition).

During the cell competition phenomenon, the interaction between the loser and the winner cells induces the activation of the apoptosis programme in the losers. Although the entire set of mechanism(s) responsible for the triggering of apoptosis is not known, a key aspect in the process is upregulation of JNK. In the wing imaginal disc, $M/+$ loser cells activate JNK prior to their elimination by apoptosis [60].

A similar process occurs during the elimination of oncogenic cells. The role of JNK has been studied in a group of genes, collectively called tumour-suppressing genes (TSG) [61]. These genes may be involved in the establishment of the apico-basal polarity of epithelial cells (*lgl, scrib, dlg*) or are components of the endosomal trafficking machinery (*rab5, Vps25, avalanche*). A common feature of mutations at TSG genes is that mutant larvae develop large tumours, affecting primarily the imaginal discs and the nervous system, indicating that the mutant cells are viable and can sustain continuous proliferation until the larva dies.

However, a large number of reports [24,57,58,62,63] have shown that in spite of their inherent viability, clones of TSG mutant cells die by JNK-mediated apoptosis when surrounded by normal cells (figure 2a,a′). Oncogenic TSG cells behave as losers of a cell competition assay and are removed from the tissue, thus revealing the tumour-suppressing role of cell competition.

The role of JNK as executor of TSG cells is illustrated by the finding that the suppression of JNK activity impedes the elimination of mutant cells for *lgl* [24] and *scrib* (figure 2b). Thus, cell competition makes use of the pro-apoptotic function of JNK to eliminate oncogenic cells.

To date, little is known about the molecular mechanism of JNK activation during cell competition. Recent work from the group of Johnston [64] on Myc-induced cell competition suggests the implication of the innate immune system in the process. The elimination of the loser cells requires the appearance of the Toll ligand Spatzle and subsequent activation of the Toll pathway, which eventually results in the activation of pro-apoptotic genes *hid* or *rpr*, mediated by the transcription factors Relish or Dl. Presumably, JNK is activated through the function of *hid* or *rpr*.

However, it is not clear whether this mechanism applies to other cases of cell competition such as Minute-induced cell competition or the elimination of oncogenic cells. For the latter, a different mechanism has been proposed by the Igaki group [65], which involves the Sas/PTP10D ligand/receptor system. During the interaction between *scribble* mutant cells and the surrounding wild-type cells, the winner cells re-localize the Sas ligand to their lateral surface, whereas the *scribble* cells re-localize there the PTP10D receptor. The activation of the PTP10D receptor in the *scribble* cells reduces EGFR signalling, which in turn leads to upregulation of JNK and subsequent cell elimination. However, this mechanism does not function in Minute-induced cell competition [65].

Regarding JNK activity during the elimination of oncogenic cells, we have found that clones of *scrib* mutant cells exhibit increased ROS activity (figure 2c), which can lead to JNK induction [43]. This kind of observation suggests that the appearance of an oncogenic cell is considered as a stress to the population. How this mechanism of ROS-mediated

royalsocietypublishing.org/journal/rsob   Open Biol. **9**: 180256

*scrib⁻ GFP clones*

*scrib⁻ puc GFP clones*

(a) GFPMmp1Dcp1 | (a') Mmp1 | (a'') Dcp1 | (b) GFPMmp1Dcp1

*scrib⁻ GFP clones*

(c) GFPDHE | (c') DHE

**Figure 2.** 'Loser' cells are eliminated by JNK activation. (*a*) Region of the wing disc showing clones of *scrib⁻* cells 72 h after induction (GFP) being eliminated by cell competition. These cells activate JNK as indicated by the presence of Mmp1 (red, *a'*), a known JNK target and are in apoptosis as shown by Dcp1 staining (blue, *a''*). Note the fragmentation of many of the clones. (*b*) Clones of *scrib⁻* cells 72 h after induction in which JNK is blocked by the expression of its negative regulator the phosphatase Puc (GFP). Note that there is no Mmp1 (red) or Dcp1 (blue) in the clones. These clones do not enter apoptosis and appear to grow normally. (*c*) A clone of *scrib⁻* cells 72 h after induction (GFP) showing the adventitious presence of ROS, indicated by the label with dihydroethidium (DHE), not seen in the rest of the disc.

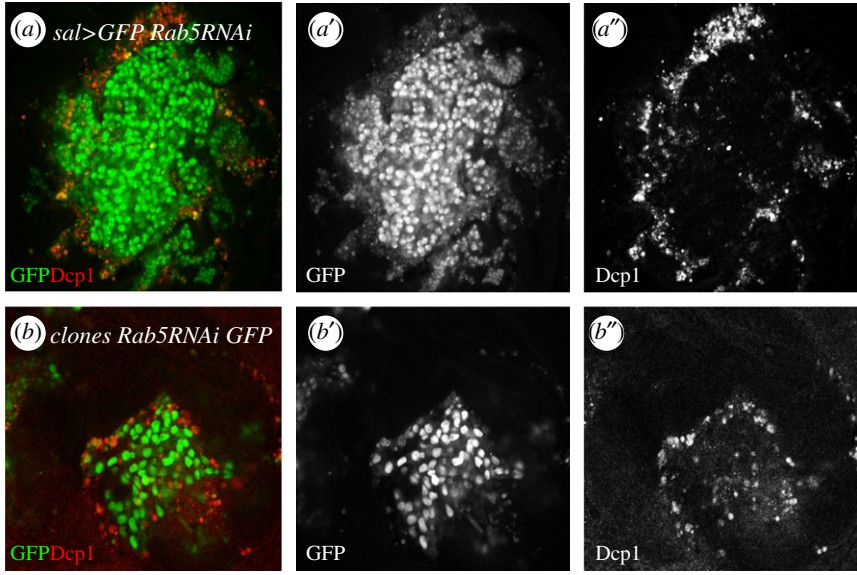

**Figure 3.** The group protection mechanism to evade cell competition. (*a,b*) Wing discs expressing Rab5 RNAi in the pouch region (*a–a'*) or in a clone (*b–b'*) labelled with GFP (green). Many of the Rab5 RNAi cells that are located at the border, in contact with surrounded wild-type cells, are in apoptosis as indicated by the Dcp1 label (red, *a''* and *b''*). The Rab5 RNAi cells inside the domain that are not in contact with wild-type cells are beyond the range of cell competition and can continue proliferating.

JNK activation relates to the Sas/PTP10D ligand/receptor system described by Yamamoto *et al.* [65] is unclear.

## 1.3. Paracrine JNK function in tumorigenesis

Sustained and inappropriate expression of JNK pathway is associated with tumour development in vertebrates and in *Drosophila* [7,25,57,58,66–68]. Regarding *Drosophila*, there are several examples in which sustained JNK activity causes tumorigenesis in imaginal discs, due to persistent proliferative signalling emanating from JNK-expressing cells [7,25,57,58,68].

The first example is the formation of tumours in the Sal domain of the wing disc by cells defective in *rab5* activity (*rab5^{KD}*). The Sal domain covers approximately 15% of the wing pouch. Isolated clones of *rab5^{KD}* are normally eliminated by JNK-mediated cell competition [58], but when the entire Sal domain is made of *rab5^{KD}* cells, many of these do not die and they proliferate even though they are surrounded by non-tumour cells. We believe that the reason behind this behaviour derives from the fact that cell competition is a short-range mechanism. The original group of *rab5^{KD}* Sal domain comprises about 400–500 cells and is approximately square. The size and shape of this domain ensures that many of the cells inside the group are beyond the reach of cell competition and therefore can continue proliferating. We have referred to this process as 'group protection' [58]: cell competition/apoptosis is mainly restricted to the border of patches deficient for *rab5* (figure 3). In this situation, tumour cells die at the border, but

are continuously replaced by neighbours, which are also subjected to cell competition and die. This process is reiterated for the rest of the development but allows persistent apoptosis and JNK activity at the tumour borders. It is the proliferative signalling emanating from JNK-expressing cells that stimulates the growth of the tumour, if JNK activity is suppressed, the growth of the tumour is also suppressed [58].

In this situation, cell competition is not sufficiently effective as to remove all the tumour cells. The continuous appearance of cells in apoptosis causes persistent proliferative signalling that stimulates mitogenic activity of tumour cells. Under these circumstances, the tandem cell competition/apoptosis reverses their anti-tumour function and becomes pro-tumorigenic [58].

Another example of the transformation of JNK activity from pro-apoptotic to pro-proliferative comes from work by Yamamoto et al. [65]. In experiments using oncogenic scribble mutant cells as an assay, these authors find that when the Sas/PTP10D cell competition system is suppressed, there is increased EGFR activity, known to prevent apoptosis in Drosophila epidermal cells [69]. It releases the pro-proliferate activity of JNK, which downregulates Hippo and gives rise to tissue overgrowth caused by over-proliferation of scribble cells.

It is of interest that there are examples from the vertebrate field also pointing to the tumour-stimulating function of apoptotic cells. It has been shown that mouse apoptotic cells secrete the cytokine Prostaglandin 2, a growth factor [70], which acts as a proliferative signal. One significant observation is that the growth of rat tumour cells is greatly enhanced when they are mixed with other lethally irradiated cells [70–73]. These observations could have some clinical implications as cancer patients are often treated with pro-apoptotic agents (X-rays, chemotherapy). In case, the treatment is not sufficiently effective, the surviving cancer cells would receive a proliferative stimulus that may lead to repopulation of the tumour.

Finally, there is a related observation [25] based on apoptosis-deficient cells that acquire sustained JNK expression. In Drosophila, but also in mammalian cells [7,74], stress treatments like X-radiation induce JNK activity, which in cells open to apoptosis leads to the demise of the cells (figure 1). However, cells unable to enter apoptosis (they may be defective in the apoptosis machinery or express high levels of the Ras pathway, which suppresses apoptosis) survive stress treatments and acquire persistent JNK activity for the rest of the development. This activity is translated into continuous function of JAK/STAT and Wg/Dpp and the formation of overgrowths (figure 4). The conclusion is that a short-term stress, which would be inconsequential in cells open to apoptosis, would cause indefinite cell proliferation of these cells. These results point to an inherent tumorigenic potential of apoptosis-deficient cells, which may respond with extra-proliferation after situations that would have little or no effect in tissues open to apoptosis.

The tumorigenic potential of apoptosis-defective cells may also be responsible for the tumorigenesis caused by deregulated expression of the Ras pathway, which is associated with the development of many different types of cancer in humans [75–78].

In Drosophila, it was observed some time ago [69,79] that constitutive activity of the Ras pathway makes cells refractory to apoptosis. In Drosophila, the overexpression of Ras (making use of the $ras^{V12}$ minigene [80]) only causes modest hyperplastic overgrowths [23–25,57,68], but after irradiation, there is

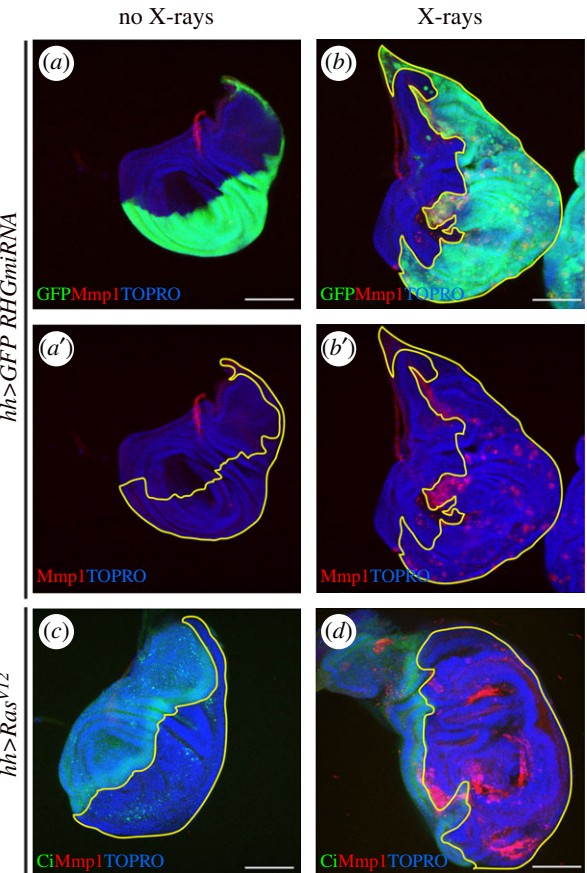

**Figure 4.** Overgrowth induced by persistent JNK activity. (a,b) Wing discs expressing in the posterior compartment the RHGmiRNA construct that supresses the activity of the pro-apoptotic genes rpr, hid and grim, labelled with GFP (green). JNK activity is monitored by staining with the matrix metaloprotease1 (Mmp1). The disc in (a,a′) has not been irradiated and shows no ectopic JNK activity nor overgrowth. The disc in (b,b′) is of the same genotype but was irradiated (3000 R) 72 h before fixation. Note the presence of numerous regions with Mmp1 expression and also the overgrowth of the posterior compartment. (c,d) Wing discs expressing the oncogenic form of Ras ($Ras^{V12}$) in the posterior compartment. The A/P border is demarcated by the expression of cubitus interruptus (ci, green) that labels the anterior compartment. The disc in (c) is not irradiated and presents neither ectopic JNK activity nor overgrowth. The disc in (d) is of the same genotype but was irradiated 72 h before fixation. Note persistent JNK activity (red) and overgrowth of the posterior compartment. The regions of interest are outlined in yellow.

persistent JNK activity and associated pathways that causes large overgrowths (figure 4c,d) [25]. These results suggest that Ras-expressing cells acquire tumorigenic potential as a consequence of the inhibition of apoptosis that allows continuous JNK activity after stress. They also suggest at least in Drosophila, the principal tumorigenic feature of the Ras pathway lies on its ability to suppress apoptosis when overexpressed.

There is the possibility that some of the oncogenic potential of Ras overexpression in vertebrates may derive from interference with apoptosis. It has been shown that Ras also has an anti-apoptotic function in mammalian cells [81] and also that the tumorigenesis associated with Ras overexpression requires JNK activity [82]. Thus, both in Drosophila and vertebrates, RAS and JNK activities may be associated with tumour development through suppression of apoptosis.

The mechanism of JNK sustenance after transient activation is of interest, because it indicates that in the absence of apoptosis the pathway is self-supporting. The finding of the

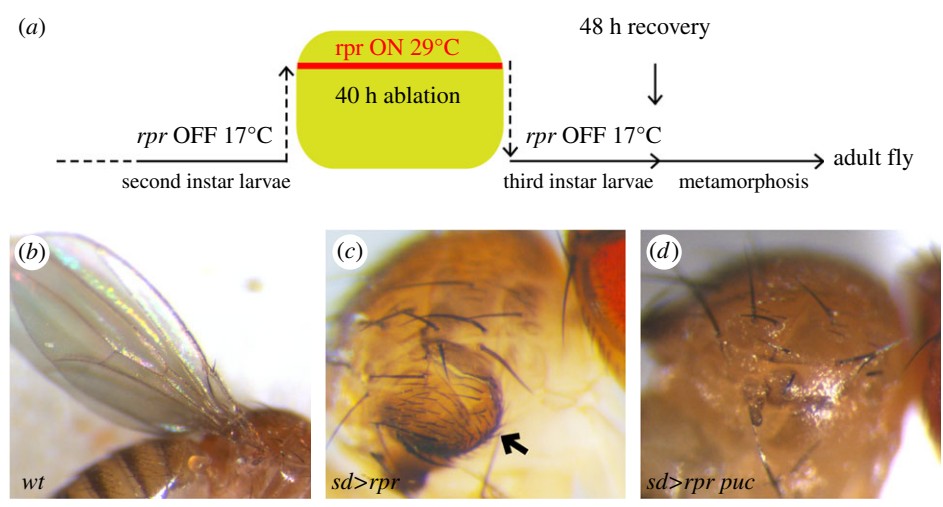

**Figure 5.** JNK implication in the regeneration process. (*a*) Scheme illustrating the ablation procedure. The *sd-Gal4* line is expressed in all the appendage cells (wing proper and hinge) of the wing disc. Larvae of the *sd-Gal4 > UAS-rpr Gal80*$^{TS}$ genotype are kept at 17°C, a temperature at which there is no *rpr* activity because the Gal80$^{TS}$ dominant suppressor of Gal4 is active at that temperature. When shifted to 29°C, the Gal80$^{TS}$ is inactivated, allowing Gal4 and *rpr* activity. In the experiments shown in the figure, the ablation period was of 40 h and started at the end of the second larval instar. A shift back to 17°C stops ablation and allows recovery for the remaining of the development. Adult flies emerge after the treatment and the effect can be inspected in the adult structures. (*b*) Photograph of a wing and notum of a wild-type fly (wt). (*c*) Photograph of an adult fly subjected during the larval period to the 40 h ablation procedure described in (*a*) and showing a notum duplication (arrow). (*d*) Photograph of an adult fly of *sd-Gal4 > UAS-rpr Gal80*$^{TS}$ *UAS-puc* genotype subjected to 40 h ablation following the same procedure. In this genotype, JNK activity is blocked, only in the appendage cells where the ablation is induced, by the expression of the phosphatase Puckered (*puc*). Note the absence of a duplicated notum. The magnification of (*b*) is lower than that of (*c*) and (*d*).

JNK target *moladietz (mol)*, a gene encoding a Duox maturation factor necessary for ROS production [27], suggests a plausible model of JNK sustenance. The original activation of JNK (generated by damage, or P53 or JNK itself) induces *mol* function, which stimulates ROS production that in turn stimulates JNK [25]. It results in an amplification loop that maintains continuous activity of JNK (figure 1*a*).

### 1.4. Paracrine JNK function in regeneration in *Drosophila*

The JNK pathway has been shown to be upregulated and required for regeneration after damage or amputation in *Drosophila* and other organisms [1,43,83–86]. As shown above, a principal property of JNK is its capacity to generate proliferative signalling, which strongly suggests that this property may be involved in generating the necessary cell proliferation during the regeneration processes.

A direct proof of the role of JNK inducing increased proliferation of regenerative cells in the wing disc has been recently reported [26]. In those experiments, the pro-apoptotic gene *reaper* (*rpr*) is activated in Scalloped domain, which includes the precursor cells of the wing and hinge (the whole appendage) but not of the notum (figure 5*a*). The appendage cells acquire high levels of apoptosis and JNK and are eventually eliminated, while there is no apoptosis/JNK in the notum cells. However, there is an increase in the proliferation of notum cells close to the dying appendage cells, associated with downregulation of the Hippo pathway. Surprisingly, over-proliferating notum cells are unable to regenerate appendage structures but form instead a duplicate of the notum (figure 5*c*, arrow). A key observation is that suppression of JNK activity in the dying cells in turn suppresses the over-proliferation of notum cells and the appearance of a notum duplication. The emerging adults contain a regular notum and no appendage (figure 5*d*). The conclusion is that the proliferative stimulus received by notum cells derives from JNK activity in dying appendage cells.

Another indication of the involvement of JNK in regeneration processes comes from the comparison of the different regenerative potential between the notum and the appendage: the latter exhibits a strong regenerative potential after damage, in contrast with the notum, which does not regenerate [26]. A reason behind this distinct likely resides on the differential function of JNK in those tissues. We have shown that, unlike in the appendage, forcing JNK activity in the notum does not induce over-proliferation [26].

The involvement of JNK in regeneration is mediated by some of its downstream signalling pathways JAK/STAT and Wg, known to function as growth factors [87–90] and appear to play an important role during regeneration. These two pathways have been shown to be implicated in regeneration of imaginal discs of *Drosophila* [90–92]. The Wnt pathway is required for regeneration in vertebrates [93–98], *Hydra* [99] and planarias [100].

## 2. General conclusion and perspectives

Just focusing on *Drosophila*, it is clear from the above that the JNK pathway, through its pro-apoptotic and pro-proliferative functions, is involved in relevant physiological processes like cell competition, tumorigenesis and regeneration. However, the genetic/molecular mechanisms behind the various roles of JNK are not well known. This is especially the case of the paracrine pro-proliferation function. This role is, at least in part, mediated by downstream pathways like JAK/STAT, Wg and Dpp. However, the relative contribution of each of these pathways is not known. A systematic study of the pro-proliferative activity of JNK in the absence of JAK/STAT, Wg or Dpp function has not been performed. It is also unclear whether those pathways form a linear cascade or act independently.

One possible way to progress may be to design screens to identify and characterize specific genes whose expression is altered in tissues that overgrow in response to sustained JNK activity.

Data accessibility. This article has no additional data.

Authors' contributions. N.P. and G.M. have elaborated the text and figures. M.C. has contributed with relevant observations about scribble mutant cells.

Competing interests. We declare we have no competing interests.

Funding. This work was supported by Ministerio de Economia y Competitividad (BFU2015-67839-P MINECO-FEDER) and Grant XVII Concurso Nacional from the Fundación Ramón Areces.

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
