## [Reviewer comments · Open Biology]

Review History

RSOB-18-0256.R0 (Original submission)

Review form: Reviewer 1

Recommendation

Accept with minor revision (please list in comments)

Are each of the following suitable for general readers?

- a) **Title**
Yes
- b) **Summary**
Yes
- c) **Introduction**
Yes

Is the length of the paper justified?

Yes

Should the paper be seen by a specialist statistical reviewer?

No

Is it clear how to make all supporting data available?

Not Applicable

Is the supplementary material necessary; and if so is it adequate and clear?

Not Applicable

Do you have any ethical concerns with this paper?

No

Comments to the Author

In this manuscript the authors discuss the opposite roles of the JNK pathway in apoptosis versus proliferation. The authors put together the state of the art and make a valuable contribution by analyzing different contexts and concluding that the outcome of the JNK differs when acts as autocrine or as a paracrine signal. The manuscript is accompanied by several images that support their conclusions. The manuscript considers a long-standing debate, which deserves much more attention for its relevance on cell competition, compensatory proliferation, tumor growth and hyperplasia, growth control and regeneration. Overall, this is a timely paper and deserves publication. There are few comments and suggestions that hopefully can help to improve the present version.

The authors describe the dual function of JNK, which acts as paracrine and autocrine signal. I am missing any possible mechanism that could explain how JNK signals reach the neighbor cells. If the same JNK has a dual function and is produced by the same source of cells, downstream signals of the JNK pathway should be activated, act on the neighbors (e.g. via jak stat or wg), and then JNK activated in that nearby cell. Alternatively, the TNF Eiger should be transcriptionally activated (somehow) and signal to neighbors. My concern is that, according to the model in figure 1, whatever is the mechanism that JNK is reaching the nearby cells, transcriptional activation is supposed to be involved. But transcriptional activation, especially after apoptosis or damage, can be very important to sustain the mechanism of repair or growth, but too slow for the initial steps. The initial responses to insults are usually fast, albeit transient, in the order of seconds or minutes. These rapid responses include pro-inflammatory signals, which are often triggered by calcium and ROS, and act upstream of protein modifications (e.g. phosphorylation of JNK). In addition, some of these rapid signals are able to spread from a source to surrounding tissue. Therefore the JNK function in proliferation could be a consequence of stressors propagated, at least in the initial steps, rather than the paracrine JNK. There are plenty of evidences that damage produces release of calcium, this calcium is propagated, NAPH oxidases (Duox) activated by calcium, and these oxidases produce H₂O₂ that activate JNK in proliferating cells. Thus a mechanism that propagates JNK could be scheduled as: Damage->Ca⁺⁺->ROS->JNK->Proliferation

which has been proposed in Drosophila discs (W. Wood, P. Martin, F. Serras, M. Gallo labs) . Note that the paracrine function would be supported by the Ca⁺⁺ module->ROS or alternatively, as proposed by some, by the ROS propagation itself. Thus, initially JNK is activated in both damaged cells and growing cells. Summarizing, the authors mention ROS and that ROS is important for JNK activation. However they omit that the mechanism of stress-dependent JNK activation could also explain the action of JNK in growing cells. It would be fair to consider this issue in the manuscript, particularly that stress can act on JNK, or at least that this stress-

dependent JNK activation could operate in conjunction with paracrine model presented here. This would fit in a comprehensive model in which JNK generated in tumor cells or apoptotic cells creates a stress environment that when propagated, cells respond by activating JNK or enhancing JNK activity.

Minor comments

Page 4. This sentence should be in the previous paragraph: 'The mechanism by which Dronc activates or stimulates JNK (37, 38) is not known'.

Page 6 first sentence: 'X-ray-damaged tissue can compensate in absence of the mitogenic signals Wg and Dpp from apoptotic cells (7).' In the way it is written one can conclude that Wg and Dpp are not involved in compensatory proliferation. This contradicts other parts of the manuscript, where the authors comment that wg is involved in growth. It has been demonstrated that Wg is a key signal for growth; an enhancer of wg is activated in tumors and regeneration in a JNK-dependent manner (Hariharan, Schubiger). Moreover, tumor growth dependent of JNK has been shown to act via Wg (Milan. Dekanty et al., 2012. Moreover, in the absence of that wg enhancer, discs do not regenerate and tumor growth diminishes. These points could be clarified.

Page 6 the sentence 'Although it was discovered in cells in which the apoptosis program was active, the paracrine capacity to stimulate cell proliferation is an intrinsic property of JNK that does not depend on apoptosis. Sustained JNK activity in apoptosis-deficient cells also causes excessive proliferation (7, 25).'

Again this could be dependent on stress signals, as mentioned above. Sustained activity of JNK could result in sustained activation of caspases, which in turn, result in high stress, affecting mitochondria and generating high oxidative stress. Can the authors discard that the short range signal (paracrine) is amplified because stress can propagate to neighbors?.

Page 6. About the sentence: 'The paracrine function is normally inconsequential because the cells in apoptosis emitting mitogenic signals are short-lived. It is only when the apoptotic cells remain alive, like the undead cells, or when cells lack the apoptosis machinery (7, 25), that the consequences of the proliferative signalling become apparent' This a good point. Indeed cell death would prevent soon or late the paracrine signal. Therefore, for compensatory proliferation, other signals might be important. Again, ROS and Ca, which are produced in dying and dead cells for long periods, could be the trigger.

Page 8 top. The sentence 'population. How this mechanism of ROSmediated JNK activation relates to the Sas/PTP10D ligand/receptor system described by Yamamoto et al is unclear' could be moved to the Sas/PTP10D paragraph below of the same page.

The 'group protection' concept is very intriguing and interesting. The figure is very convincing. In fact, it can be considered a variation of the Gurdon's community effect; why not using the term community effect?

The JNK sustenance by mol is a mechanism to maintain the stress, but not to originate it. ROS are primarily induced in the mitochondria. Again, ROS are produced far earlier than mol expression.

Figure 2c Clarify in the legend whether those clones, in addition to scrib-, overexpress the phosphatase puc or are visualizing puc-GFP. This could help to general readers. Is the scale the same as b?

Page 9 last paragraph of the regeneration section. The reference Mattila et al., is mainly related to Dpp.

Scale bars missing

Figure 5 Instead 'the interphase 2/3 larval period' consider using 'between the second and third larval phases' or 'at the end of the 2nd larval instar' as in the figure.

Review form: Reviewer 2

Recommendation

Accept with minor revision (please list in comments)

Are each of the following suitable for general readers?

- a) **Title**
Yes
- b) **Summary**
Yes
- c) **Introduction**
Yes

Is the length of the paper justified?

Yes

Should the paper be seen by a specialist statistical reviewer?

No

Is it clear how to make all supporting data available?

Not Applicable

Is the supplementary material necessary; and if so is it adequate and clear?

Not Applicable

Do you have any ethical concerns with this paper?

No

Comments to the Author

The JNK signaling pathway is an important pathway conserved among metazoans. It plays an important role in different processes such as tissue homeostasis, morphogenesis, wound healing, immunity, programmed cell death or aging. Therefore, it presents great interest for a large panel of readers. In this review paper, the authors present the complexity of the JNK functions in vivo with respect to cell proliferation, competition, tumorigenesis and regeneration. Most of the paper deals with data obtained using the *Drosophila* wing imaginal disc model, but the authors also open up to the mammalian field. At the end of the review, the authors present very novel data that compare regeneration processes in two different tissues, the notum and the appendage. They suggest that differences between models are mediated by differing downstream signaling pathways. I agree with the authors but it could also be partly explained by the large number of possible kinases, adaptors and ligand/receptor combinations.

Overall, this is a nice manuscript that can be published with minor revisions (see pdf file).

Decision letter (RSOB-18-0256.R0)

04-Feb-2019

Dear Professor Morata

We are pleased to inform you that your manuscript RSOB-18-0256 entitled "Pro-apoptotic and pro-proliferation functions of the JNK pathway of *Drosophila*: roles in cell competition, tumorigenesis and regeneration" has been accepted by the Editor for publication in *Open Biology*. The reviewer(s) have recommended publication, but also suggest some minor revisions to your manuscript. Therefore, we invite you to respond to the reviewer(s)' comments and revise your manuscript.

Please submit the revised version of your manuscript within 14 days. If you do not think you will be able to meet this date please let us know immediately and we can extend this deadline for you.

- 1) A text file of the manuscript (doc, txt, rtf or tex), including the references, tables (including captions) and figure captions. Please remove any tracked changes from the text before submission. PDF files are not an accepted format for the "Main Document".
- 2) A separate electronic file of each figure (tiff, EPS or print-quality PDF preferred). The format should be produced directly from original creation package, or original software format. Please note that PowerPoint files are not accepted.
- 3) Electronic supplementary material: this should be contained in a separate file from the main text and meet our ESM criteria (see <http://royalsocietypublishing.org/instructions-authors#question5>). All supplementary materials accompanying an accepted article will be treated as in their final form. They will be published alongside the paper on the journal website and posted on the online figshare repository. Files on figshare will be made available approximately one week before the accompanying article so that the supplementary material can be attributed a unique DOI.

Online supplementary material will also carry the title and description provided during

submission, so please ensure these are accurate and informative. Note that the Royal Society will not edit or typeset supplementary material and it will be hosted as provided. Please ensure that the supplementary material includes the paper details (authors, title, journal name, article DOI). Your article DOI will be 10.1098/rsob.2016[last 4 digits of e.g. 10.1098/rsob.20160049].

4) A media summary: a short non-technical summary (up to 100 words) of the key findings/importance of your manuscript. Please try to write in simple English, avoid jargon, explain the importance of the topic, outline the main implications and describe why this topic is newsworthy.

Images

Data-Sharing

It is a condition of publication that data supporting your paper are made available. Data should be made available either in the electronic supplementary material or through an appropriate repository. Details of how to access data should be included in your paper. Please see <http://royalsocietypublishing.org/site/authors/policy.xhtml#question6> for more details.

Data accessibility section

Sincerely,

The Open Biology Team
<mailto:openbiology@royalsociety.org>

Reviewer(s)' Comments to Author:

Referee: 1

Comments to the Author(s)

In this manuscript the authors discuss the opposite roles of the JNK pathway in apoptosis versus proliferation. The authors put together the state of the art and make a valuable contribution by analyzing different contexts and concluding that the outcome of the JNK differs when acts as autocrine or as a paracrine signal. The manuscript is accompanied by several images that support their conclusions. The manuscript considers a long-standing debate, which deserves much more attention for its relevance on cell competition, compensatory proliferation, tumor growth and hyperplasia, growth control and regeneration. Overall, this is a timely paper and deserves publication. There are few comments and suggestions that hopefully can help to improve the present version.

The authors describe the dual function of JNK, which acts as paracrine and autocrine signal. I am missing any possible mechanism that could explain how JNK signals reach the neighbor cells. If the same JNK has a dual function and is produced by the same source of cells, downstream signals of the JNK pathway should be activated, act on the neighbors (e.g. via jak stat or wg), and then JNK activated in that nearby cell. Alternatively, the TNF Eiger should be transcriptionally activated (somehow) and signal to neighbors. My concern is that, according to the model in figure 1, whatever is the mechanism that JNK is reaching the nearby cells, transcriptional activation is supposed to be involved. But transcriptional activation, especially after apoptosis or damage, can be very important to sustain the mechanism of repair or growth, but too slow for the initial steps. The initial responses to insults are usually fast, albeit transient, in the order of seconds or minutes. These rapid responses include pro-inflammatory signals, which are often triggered by calcium and ROS, and act upstream of protein modifications (e.g. phosphorylation of JNK). In addition, some of these rapid signals are able to spread from a source to surrounding tissue. Therefore the JNK function in proliferation could be a consequence of stressors propagated, at least in the initial steps, rather than the paracrine JNK. There are plenty of evidences that damage produces release of calcium, this calcium is propagated, NAPH oxidases (Duox) activated by calcium, and these oxidases produce H₂O₂ that activate JNK in proliferating cells. Thus a mechanism that propagates JNK could be scheduled as: Damage->Ca⁺⁺->ROS->JNK->Proliferation which has been proposed in Drosophila discs (W. Wood, P. Martin, F. Serras, M. Gallo labs) . Note that the paracrine function would be supported by the Ca⁺⁺ module->ROS or alternatively, as proposed by some, by the ROS propagation itself. Thus, initially JNK is activated in both damaged cells and growing cells. Summarizing, the authors mention ROS and that ROS is important for JNK activation. However they omit that the mechanism of stress-dependent JNK activation could also explain the action of JNK in growing cells. It would be fair to consider this issue in the manuscript, particularly that stress can act on JNK, or at least that this stress-dependent JNK activation could operate in conjunction with paracrine model presented here. This would fit in a comprehensive model in which JNK generated in tumor cells or apoptotic cells creates a stress environment that when propagated, cells respond by activating JNK or enhancing JNK activity.

Minor comments

Page 4. This sentence should be in the previous paragraph: 'The mechanism by which Dronc activates or stimulates JNK (37, 38) is not known'.

Page 6 first sentence: ' X-ray-damaged tissue can compensate in absence of the mitogenic signals Wg and Dpp from apoptotic cells (7).' In the way it is written one can conclude that Wg and Dpp are not involved in compensatory proliferation. This contradicts other parts of the manuscript, where the authors comment that wg is involved in growth. It has been demonstrated that Wg is a key signal for growth; an enhancer of wg is activated in tumors and regeneration in a JNK-dependent manner (Hariharan, Schubiger). Moreover, tumor growth dependent of JNK has been shown to act via Wg (Milan. Dekanty et al., 2012. Moreover, in the absence of that wg enhancer, discs do not regenerate and tumor growth diminishes. These points could be clarified.

Page 6 the sentence 'Although it was discovered in cells in which the apoptosis program was active, the paracrine capacity to stimulate cell proliferation is an intrinsic property of JNK that does not depend on apoptosis. Sustained JNK activity in apoptosis-deficient cells also causes excessive proliferation (7, 25).'

Again this could be dependent on stress signals, as mentioned above. Sustained activity of JNK could result in sustained activation of caspases, which in turn, result in high stress, affecting mitochondria and generating high oxidative stress. Can the authors discard that the short range signal (paracrine) is amplified because stress can propagate to neighbors?.

Page 6. About the sentence: 'The paracrine function is normally inconsequential because the cells in apoptosis emitting mitogenic signals are short-lived. It is only when the apoptotic cells remain alive, like the undead cells, or when cells lack the apoptosis machinery (7, 25), that the consequences of the proliferative signalling become apparent'

This a good point. Indeed cell death would prevent soon or late the paracrine signal. Therefore, for compensatory proliferation, other signals might be important. Again, ROS and Ca, which are produced in dying and dead cells for long periods, could be the trigger.

Page 8 top. The sentence 'population. How this mechanism of ROSmediated JNK activation relates to the Sas/PTP10D ligand/receptor system described by Yamamoto et al is unclear' could be moved to the Sas/PTP10D paragraph below of the same page.

The 'group protection' concept is very intriguing and interesting. The figure is very convincing. In fact, it can be considered a variation of the Gurdon's community effect; why not using the term community effect?

The JNK sustenance by mol is a mechanism to maintain the stress, but not to originate it. ROS are primarily induced in the mitochondria. Again, ROS are produced far earlier than mol expression.

Figure 2c Clarify in the legend whether those clones, in addition to scrib-, overexpress the phosphatase puc or are visualizing puc-GFP. This could help to general readers. Is the scale the same as b?

Page 9 last paragraph of the regeneration section. The reference Mattila et al., is mainly related to Dpp.

Scale bars missing

Figure 5 Instead 'the interphase 2/3 larval period' consider using 'between the second and third larval phases' or 'at the end of the 2nd larval instar' as in the figure.

Referee: 2

Comments to the Author(s)

The JNK signaling pathway is an important pathway conserved among metazoans. It plays an important role in different processes such as tissue homeostasis, morphogenesis, wound healing, immunity, programmed cell death or aging. Therefore, it presents great interest for a large panel of readers. In this review paper, the authors present the complexity of the JNK functions in vivo with respect to cell proliferation, competition, tumorigenesis and regeneration. Most of the paper deals with data obtained using the Drosophila wing imaginal disc model, but the authors also open up to the mammalian field. At the end of the review, the authors present very novel data that compare regeneration processes in two different tissues, the notum and the appendage. They suggest that differences between models are mediated by differing downstream signaling pathways. I agree with the authors but it could also be partly explained by the large number of possible kinases, adaptors and ligand/receptor combinations.

Overall, this is a nice manuscript that can be published with minor revisions (see pdf file).

Author's Response to Decision Letter for (RSOB-18-0256.R0)

See Appendix A.

Decision letter (RSOB-18-0256.R1)

18-Feb-2019

Dear Professor Morata,

We are pleased to inform you that your manuscript entitled "Pro-apoptotic and pro-proliferation functions of the JNK pathway of *Drosophila*: roles in cell competition, tumorigenesis and regeneration" has been accepted by the Editor for publication in Open Biology.

Sincerely,

The Open Biology Team
mailto: openbiology@royalsociety.org

Appendix A

Response to referees

We would like to thank the two referees for their positive and constructive comments. They appreciate the value of our manuscript in putting together the evidence indicating the dual functions of JNK regarding apoptosis and cell proliferation.

Both of them point out some grammatical mistakes and also some sentences that are not sufficiently clear. They suggest appropriate alterations that we have included in the revised version of the manuscript

In the case of Referee 1, this person raises several points. A major one concerning our model that JNK may exert two distinct functions: an autocrine cell killing activity in the cells expressing the pathway and a paracrine pro-proliferation function affecting neighbour cells. The referee argues about the possibility of the paracrine signalling emanating from the JNK expressing cells would result in JNK activation in the neighbour cells caused by stressors (calcium, H₂O₂). This JNK function might be responsible of the pro-proliferation effect. We are aware of work indicating the JNK activity may be propagated, via eiger or other factors. However, the important issue is that it is the autocrine cell-killing function of JNK what causes proliferation of neighbour cells. Whether this is mediated by secondary non-lethal activation of JNK is of interest but does not affect the essence of the model: cells expressing JNK undergo cell death themselves (autocrine effect), and also generate signals that stimulate proliferation of neighbour cells - a paracrine effect. Perhaps the best demonstration of this duality comes from the regeneration experiments reported by Martin et al Development 2017, which are also illustrated in Figure 5c,d of our manuscript. rpr-mediated killing of the wing cells causes over-proliferation of their neighbour notum cells, which make a notum duplicate. The over-proliferation and the notum duplication are suppressed if the dying wing cells cannot activate JNK. Thus it is the signalling emanating from JNK-expressing dying cells that is responsible for the overgrowth of neighbour tissue

The referee comments about a sentence of our manuscript "*X-ray-damaged tissue can compensate in absence of the mitogenic signals Wg and Dpp from apoptotic cells*" and states "*In the way it is written one can conclude that Wg and Dpp are not involved in compensatory proliferation*" That is exactly what we wanted to conclude; that Wg and

Dpp are not involved in compensatory proliferations. Strong evidence for this claim was published some years ago (Perez-Garijo et al Development 2009, cited in the reference list). Wg is certainly involved in generating overgrowths but not in compensating for lost tissue after irradiation.

The referee also points out our sentence *"Although it was discovered in cells in which the apoptosis program was active, the paracrine capacity to stimulate cell proliferation is an intrinsic property of JNK that does not depend on apoptosis. Sustained JNK activity in apoptosis-deficient cells also causes excessive proliferation"*

and then states

"Again this could be dependent on stress signals, as mentioned above. Sustained activity of JNK could result in sustained activation of caspases, which in turn, result in high stress, affecting mitochondria and generating high oxidative stress. Can the authors discard that the short range signal (paracrine) is amplified because stress can propagate to neighbors?"

The referee is in error here for in apoptosis-deficient cells there is no caspase activity

About the "group protection" concept. As postulated by John Gurdon, it describes a situation in which cells can influence the determination of their neighbours. It is a very interesting idea that we have postulated may function during regeneration (Herrera and Morata eLife 2014). However the group protection hypothesis describes a mechanism by which a sufficiently large group of tumour cells can escape cell competition because the latter is a short-range phenomenon that only affects the periphery of the group. Those inside are beyond the reach